# TOWARDS PHYSICS-INFORMED DEEP LEARNING FOR TURBULENT FLOW PREDICTION

## ABSTRACT

While deep learning has shown tremendous success in a wide range of domains, it remains a grand challenge to incorporate physical principles in a systematic manner to the design, training and inference of such models. In this paper, we aim to predict turbulent flow by learning its highly nonlinear dynamics from spatiotemporal velocity fields of large-scale fluid flow simulations of relevance to turbulence modeling and climate modeling. We adopt a hybrid approach by marrying two well-established turbulent flow simulation techniques with deep learning. Specifically, we introduce trainable spectral filters in a coupled model of Reynolds-averaged Navier-Stokes (RANS) and Large Eddy Simulation (LES), followed by a specialized U-net for prediction. Our approach, which we call Turbulent-Flow Net (`TF-Net`), is grounded in a principled physics model, yet offers the flexibility of learned representations. We compare our model, `TF-Net`, with state-of-the-art baselines and observe significant reductions in error for predictions 60 frames ahead. Most importantly, our method predicts physical fields that obey desirable physical characteristics, such as conservation of mass, whilst faithfully emulating the turbulent kinetic energy field and spectrum, which are critical for accurate prediction of turbulent flows.

## 1 INTRODUCTION

Modeling the dynamics of physical processes that evolve over space and time and vary over a wide range of spatial and temporal scales is a fundamental task in science. Computational fluid dynamics (CFD) is at the heart of climate modeling and has direct implications for understanding and predicting climate change. However, the current paradigm in atmospheric CFD is purely *physics-driven*: known physical laws encoded in systems of coupled partial differential equations (PDEs) are solved over space and time via numerical differentiation and integration schemes. These methods are tremendously computationally-intensive, requiring significant computational resources and expertise. Recently, *data-driven* methods, including deep learning, have demonstrated great success in the automation, acceleration, and streamlining of highly compute-intensive workflows for science (Reichstein et al., 2019). But existing deep learning methods are mainly statistical with little or no underlying physical knowledge incorporated, and are yet to be proven to be successful in capturing and predicting accurately the properties of complex physical systems.

Developing deep learning methods that can incorporate physical laws in a systematic manner is a key element in advancing AI for physical sciences (Steven Brunton, 2019). Towards this goal, we investigate the challenging problem of predicting a turbulent flow, governed by the high-dimensional non-linear Navier-Stokes equations. Recently, several studies have attempted incorporating knowledge about a physical system into deep learning. For example, Emmanuel de Bezenac (2018) proposed a warping scheme to predict the sea surface temperature, but only considered the linear advection-diffusion equation. Xie et al. (2018) and Jonathan Tompson (2017) developed deep learning models in the context of fluid flow animation, where physical consistency is less critical. Wu et al. (2019) and Tom Beucler (2019) introduced statistical and physical constraints in the loss function to regularize the predictions of the model. However, their studies only focused on spatial modeling without temporal dynamics, besides regularization being ad-hoc and difficult to tune the hyper-parameters.

In this work, we propose a hybrid learning paradigm that unifies turbulence modeling and deep representation learning. We develop a novel deep learning model, Turbulent-Flow Net (`TF-Net`), that en-

hances the capability of predicting complex turbulent flows with deep neural networks. `TF-Net` applies scale separation to model different ranges of scales of the turbulent flow individually. Building upon a promising and popular CFD technique, the RANS-LES coupling approach (E. Labourasse, 2004), our model replaces *a priori* spectral filters with trainable convolutional layers. We decompose the turbulent flow into three components, each of which is approximated by a specialized U-net to preserve invariance properties. To the best of our knowledge, this is the first hybrid framework of its kind for predicting turbulent flow. We compare our method with state-of-the-art baselines for forecasting velocity fields up to 60 steps ahead given the history. We observe that `TF-Net` is capable of generating accurate and physically meaningful predictions that preserve critical quantities of relevance. In summary, our contributions are as follows:

1. We study the challenging task of turbulent flow prediction as a test bed to investigate incorporating physics knowledge into deep learning in a principled fashion.

2. We propose a novel hybrid learning framework, `TF-Net`, that unifies a popular CFD technique, RANS-LES coupling, with custom-designed deep neural networks.

3. When evaluated on turbulence simulations, `TF-Net` achieves 11.1% reduction in prediction RMSE, 30.1% improvement in the energy spectrum, 21% turbulence kinetic energy RMSEs and 64.2% reduction of flow divergence in difference from the target, compared to the best baseline.

## 2 BACKGROUND IN TURBULENCE MODELING

Most fluid flows in nature are turbulent, but theoretical understanding of solutions to the governing equations, the Navier–Stokes equations, is incomplete. Turbulent fluctuations occur over a wide range of length and time scales with high correlations between these scales. Turbulent flows are characterized by chaotic motions and intermittency, which are difficult to predict.

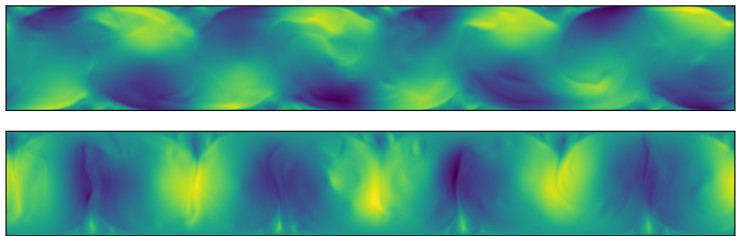

Figure 1: A snapshot of the Rayleigh-Bénard convection flow, the velocity fields along $x$ direction (top) and $y$ direction (bottom) (Chirila, 2018). The spatial resolution is 1792 x 256 pixels.

The physical system we investigate is two-dimensional Rayleigh-Bénard convection (RBC), a model for turbulent convection, with a horizontal layer of fluid heated from below so that the lower surface is at a higher temperature than the upper surface. Turbulent convection is a major feature of the dynamics of the oceans, the atmosphere, as well as engineering and industrial processes, which has motivated numerous experimental and theoretical studies for many years. The RBC system serves as an idealized model for turbulent convection that exhibits the full range of dynamics of turbulent convection for sufficiently large temperature gradients.

Let $\boldsymbol{w}$ be the vector velocity field of the flow with two components $(u, v)$, velocities along $x$ and $y$ directions, the governing equations for this physical system are:

$$\nabla \cdot \boldsymbol{w} = 0 \qquad \text{Continuity Equation}$$

$$\frac{\partial \boldsymbol{w}}{\partial t} + (\boldsymbol{w} \cdot \nabla)\boldsymbol{w} = -\frac{1}{\rho_0}\nabla p + \nu\nabla^2\boldsymbol{w} + f \qquad \text{Momentum Equation}$$

$$\frac{\partial T}{\partial t} + (\boldsymbol{w} \cdot \nabla)T = \kappa\nabla^2 T \qquad \text{Temperature Equation} \qquad (1)$$

where $p$ and $T$ are pressure and temperature respectively, $\kappa$ is the coefficient of heat conductivity, $\rho_0$ is density at temperature at the beginning, $\alpha$ is the coefficient of thermal expansion, $\nu$ is the kinematic viscosity, $f$ the body force that is due to gravity. In this work, we use a particular approach

to modeling RBC that uses a Boussinesq approximation, resulting in a divergence-free flow, so $\nabla \cdot \boldsymbol{w}$ should be zero everywhere (Chirila, 2018). Figure 1 shows a snapshot in our RBC flow dataset.

CFD allows simulating complex turbulent flows, however, the wide range of scales makes it very challenging to accurately resolve all the scales. More precisely, fully resolving a complex turbulent flow numerically, known as direct numerical simulations – DNS, requires a very fine discretization of space-time, which makes the computation prohibitive even with advanced high-performance computing. Hence most CFD methods, like Reynolds-Averaged Navier-Stokes and Large Eddy Simulations (McDonough, 2007a; Pierre Sagaut, 2006; McDonough, 2007b), resort to resolving the large scales whilst modeling the small scales, using various averaging techniques and/or low-pass filtering of the governing equations (Eqn. 1). However, the unresolved processes and their interactions with the resolved scales are extremely challenging to model. CFD remains computationally expensive despite decades of advancements in turbulence modeling and HPC.

Deep learning (DL) is poised to accelerate and improve turbulent flow simulations because well-trained DL models can generate realistic instantaneous flow fields with physically accurate spatiotemporal coherence, without solving the complex nonlinear coupled PDEs that govern the system (Tompson et al., 2017; Maziar Raissi, 2019; 2018). However, DL models are hard to train and are often used as "black boxes" in physical science as they lack knowledge of the underlying physics and are very hard to interpret. While these DL models may achieve low prediction errors they often lack scientific consistency and do not respect the physics of the systems they model. Therefore, it is critical to infusing known physics and design efficient turbulent flow prediction DL models that are not only accurate but also physically meaningful.

## 3 TURBULENT-FLOW NET

Inspired by techniques used in CFD to separate scales of this multi-scale system, the global idea behind `TF-Net` is to decompose the flow into three components of different scales with trainable modules for simulating each component. First, we provide a brief introduction of the CFD techniques which are built on this basic idea.

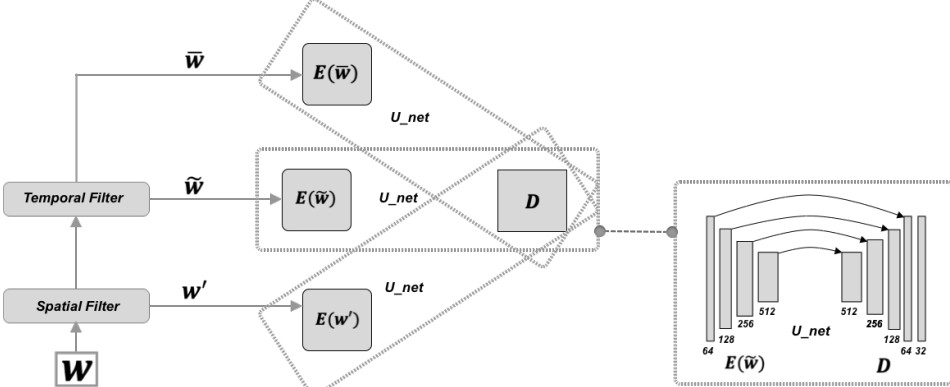

Figure 2: Turbulent Flow Net: three identical encoders to learn the transformations of the three components of different scales, and one shared decoder that learns the interactions among these three components to generate the predicted 2D velocity field at the next instant. Each encoder-decoder pair can be viewed as a U-net and the aggregation is weighted summation.

**Reynolds-averaged Navier–Stokes (RANS)** decomposes the turbulent flow $\boldsymbol{w}$ into two separable time scales: a time-averaged mean flow $\bar{\boldsymbol{w}}$ and a fluctuating quantity $\boldsymbol{w}'$. The resulting RANS equations contain a closure term, the Reynolds stresses, that require modeling, the classic closure problem of turbulence modeling. While this approach is a good first approximation to solving a turbulent flow, RANS does not account for broadband unsteadiness and intermittency, characteristic of most turbulent flows. Further, closure models for the unresolved scales are often inadequate, making RANS solutions to be less accurate. $T$ here is the moving average window size.

$$\boldsymbol{w}(\boldsymbol{x},t) = \bar{\boldsymbol{w}}(\boldsymbol{x},t) + \boldsymbol{w}'(\boldsymbol{x},t), \quad \text{where } \bar{\boldsymbol{w}}(\boldsymbol{x},t) = \frac{1}{T}\int_{t-T}^{t} G(s)\boldsymbol{w}(\boldsymbol{x},s)ds \qquad (2)$$

**Large Eddy Simulation (LES)** is an alternative approach based on low-pass filtering of the Navier-Stokes equations that solves a part of the multi-scale turbulent flow corresponding to the most energetic scales. In LES, the large scales are a spatially filtered variable $\tilde{\boldsymbol{w}}$, which is usually expressed as a convolution product by the filter kernel $G$. The kernel $G$ is often taken to be a Gaussian kernel. $\Omega_i$ is a subdomain of the solution and depends on the filter size (Sagaut, 2001).

$$\boldsymbol{w}(\boldsymbol{x},t) = \tilde{\boldsymbol{w}}(\boldsymbol{x},t) + \boldsymbol{w}'(\boldsymbol{x},t), \quad \text{where } \tilde{\boldsymbol{w}}(\boldsymbol{x},t) = \int_{\Omega_i} G(\boldsymbol{x}|\boldsymbol{\xi})\boldsymbol{w}(\boldsymbol{\xi},t)d\boldsymbol{\xi} \tag{3}$$

The key difference between RANS and LES is that RANS is based on time averaging, leading to simpler steady equations, whereas LES is based on a spatial filtering process which is more accurate but also computationally more expensive.

**Hybrid RANS-LES Coupling** combines both RANS and LES approaches in order to be able to take advantage of both methods (E. Labourasse, 2004; Chaoua, 2017). It decomposes the flow variables into three parts: mean flow, resolved fluctuations and unresolved (subgrid) fluctuations. RANS-LES coupling applies the spatial filtering operator $G_1$ and the temporal average operator $G_2$ sequentially. We can define $\bar{\boldsymbol{w}}$ in discrete form with using $\boldsymbol{w}^*$ as an intermediate term,

$$\boldsymbol{w}^*(\boldsymbol{x},\boldsymbol{t}) = G_1(\boldsymbol{w}) = \sum_{\boldsymbol{\xi}} G_1(\boldsymbol{x}|\boldsymbol{\xi})\boldsymbol{w}(\boldsymbol{\xi},t) \tag{4}$$

$$\bar{\boldsymbol{w}}(\boldsymbol{x},\boldsymbol{t}) = G_2(\boldsymbol{w}^*) = \frac{1}{T}\sum_{s=t-T}^{t} G_2(s)\boldsymbol{w}^*(\boldsymbol{x},s) \tag{5}$$

then $\tilde{\boldsymbol{w}}$ can be defined as the difference between $\boldsymbol{w}^*$ and $\bar{\boldsymbol{w}}$:

$$\tilde{\boldsymbol{w}} = \boldsymbol{w}^* - \bar{\boldsymbol{w}}, \qquad \boldsymbol{w}' = \boldsymbol{w} - \boldsymbol{w}^* \tag{6}$$

Finally we can have the three-level decomposition of the velocity field.

$$\boldsymbol{w} = \bar{\boldsymbol{w}} + \tilde{\boldsymbol{w}} + \boldsymbol{w}' \tag{7}$$

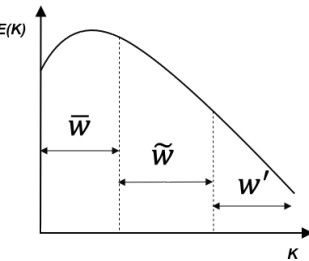

Figure 3: Three level spectral decomposition of velocity $\boldsymbol{w}$, $E(k)$ is the energy spectrum and $k$ is wavenumber.

Figure 3 shows this three-level decomposition in wavenumber space (E. Labourasse, 2004). $k$ is the wavenumber, the spatial frequency in the Fourier domain. $E(k)$ is the energy spectrum describing how much kinetic energy is contained in eddies with wavenumber $k$. Small $k$ corresponds to large eddies that contain most of the energy. The slope of the spectrum is negative and indicates the transfer of energy from large scales of motion to the small scales. This hybrid approach combines the ease and computational efficiency of RANS with the resolving power of LES to provide a technique that is less expensive and more tractable than pure LES.

**Turbulent Flow Net** We describe `TF-Net`, a hybrid deep learning framework based on the multi-level spectral decomposition of hybrid RANS-LES Coupling method. We decompose the velocity field into three components of different scales using two scale separation operators, the spatial filter $G_1$ and the temporal filter $G_2$. In traditional CFD, these filters are usually pre-defined, such as the Gaussian spatial filter. In our model, both filters are trainable neural networks. The spatial filtering process is realized by applying one convolutional layer with a single 5×5 filter to each input image. The temporal filter is implemented as a convolutional layer with a single 1×1 filter applied to every $T$ images. The motivation for this design is to explicitly guide the DL model to learn the non-linear dynamics of both large and small eddies as relevant to the task of spatio-temporal prediction.

We design three identical encoders to encode the three scale components separately. We use a shared decoder to learn the interactions among these three components and generate the final prediction. Each encoder and the decoder can be viewed as a U-net without duplicate layers and middle layer in the original architecture (Olaf Ronneberger, 2015). The encoder consists of four convolutional layers with double the number of feature channels of the previous layer and stride 2 for down-sampling. The decoder consists of one output layer and four deconvolutional layers with summation of the corresponding feature channels from the three encoders and the output of the previous layer as input. Figure 2 shows the overall architecture of our hybrid model `TF-Net`. To generate multiple time-step forecasts, we perform one-step ahead prediction and roll out autoregressively. Furthermore, since the turbulent flow under investigation has zero divergence ($\nabla \cdot \boldsymbol{w}$ should be zero everywhere), we include $||\nabla \cdot \boldsymbol{w}||^2$ as a regularizer to constrain the predictions, leading to `Con TF-Net`.

## 4 RELATED WORK

**Turbulence Modeling** Recently, machine learning models, especially DL models have been used to accelerate and improve the simulation of turbulent flows. For example, Ling et al. (2016); Fang et al. (2018) studied tensor invariant neural networks to learn the Reynolds stress tensor while preserving Galilean invariance, but Galilean invariance only applies to flows without external forces. In our case, RBC flow has gravity as an external force. Most recently, Kim & Lee (2019) studied unsupervised generative modeling of turbulent flows but the model is not able to make real time future predictions given the historic data. Raissi et al. (2017) applied a Galerkin finite element method with deep neural networks to solve PDEs automatically, what they call "Physics-informed deep learning". Though these methods have shown the ability of deep learning in solving PDEs directly and deriving generalizable solutions, the key limitation of these approaches is that they require explicitly inputs of boundary conditions during inference, which are generally not available in real-time. Arvind Mohan (2019) proposed a purely data-driven DL model for turbulence, compressed convolutional LSTM, but the model lacks physical constraints and interpretability. Wu et al. (2019) and Tom Beucler (2019) introduced statistical and physical constraints in the loss function to regularize the predictions of the model. However, their studies only focused on spatial modeling without temporal dynamics, besides regularization being ad-hoc and difficult to tune the hyper-parameters.

**Fluid Animation** In parallel, the computer graphics community has also investigated using deep learning to speed up numerical simulations for generating realistic animations of fluids such as water and smoke. For example, Tompson et al. (2017) used an incompressible Euler's equation with a customized Convolutional Neural Network (CNN) to predict velocity update within a finite difference method solver. Chu & Thuerey (2017) propose double CNN networks to synthesize high-resolution flow simulation based on reusable space-time regions. Xie et al. (2018) and Jonathan Tompson (2017) developed deep learning models in the context of fluid flow animation, where physical consistency is less critical. Steffen Wiewel (2019) proposed a method for the data-driven inference of temporal evolutions of physical functions with deep learning. However, fluid animation emphases on the realism of the simulation rather than the physical consistency of the predictions or physics metrics and diagnostics of relevance to scientists.

**Video Prediction** Our work is also related to future video prediction. Conditioning on the observed frames, video prediction models are trained to predict future frames, e.g., Mathieu et al. (2015); Finn et al. (2016); Xue et al. (2016); Villegas et al. (2017); Chelsea Finn (2016). Many of these models are trained on natural videos with complex noisy data from unknown physical processes. Therefore, it is difficult to explicitly incorporate physical principles into the model. The turbulent flow problem studied in this work is substantially different from natural video prediction because it does not attempt to predict object or camera motions. Instead, our approach aims to emulate numerical simulations given noiseless observations from known governing equations. Hence, some of these techniques are perhaps under-suited for our application.

## 5 EXPERIMENTS

### 5.1 DATASET

The dataset for our experiments comes from two dimensional turbulent flow simulated using the Lattice Boltzmann Method (Chirila, 2018). We use only the velocity vector fields, where the spatial resolution of each image is 1792 x 256. Each image has two channels, one is the turbulent flow velocity along $x$ direction and the other one is the velocity along $y$ direction. The physics parameters relevant to this numerical simulation are: Prandtl number $= 0.71$, Rayleigh number $= 2.5 \times 10^8$ and the maximum Mach number $= 0.1$. We use 1500 images (snapshots in time) for our experiments. The task is to predict the spatiotemporal velocity fields up to 60 steps ahead given 10 initial frames.

We divided each 1792 by 256 image into 7 square sub-regions of size 256 x 256, then downsample them into 64 x 64 pixels sized images. We use a sliding window approach to generate 9,870 samples of sequences of velocity fields: 6,000 training samples, 1,700 validation samples and 2,170 test samples. The DL model is trained using back-propagation through prediction errors accumulated over multiple steps. We use a validation set for hyper-parameters tuning based on the average error

of predictions up to six steps ahead. The hyper-parameters tuning range can be found in Table 2 in the appendix. All results are averaged over three runs.

## 5.2 BASELINE

We compare our model with a series of state-of-the-art baselines for turbulent flow prediction.

- `ResNet` (Kaiming He, 2015): a 34-layer Residual Network by replacing the final dense layer with a convolutional layer with two output channels.
- `ConvLSTM` (Xingjian Shi, 2015): a 3-layer Convolutional LSTM model used for spatiotemporal precipitation nowcasting.
- `U-Net` (Olaf Ronneberger, 2015): Convolutional neural networks originally developed for image segmentation, also used for video prediction.
- `GAN`: U-net trained with a discriminator like the Generative Neural Networks.
- `SST` (Emmanuel de Bezenac, 2018): hybrid deep learning model using warping scheme for linear energy equation to predict sea surface temperature, which is also applicable to the linearized momentum equation that governs the velocity fields.
- `DHPM` (Raissi, 2018): Deep Hidden Physics Model is to directly approximate the solution of partial differential equations with fully connected networks using space and time as inputs. The model is trained twice on the training set and the test set with boundary conditions.

Here `ResNet`, `ConvLSTM`, `U-net` and `GAN` are pure data-driven spatiotemporal deep learning models for video predictions. `SST` and `DHPM` are hybrid techniques that aim to incorporate prior physical knowledge into deep learning for fluid simulation.

## 5.3 EVALUATION METRICS

Even though Root Mean Square Error (RMSE) is a widely accepted metric for quantifying the differences between model predictions and the ground truth, it is still insufficient to apply the predicted turbulent flows with good RMSE to scientific fields. We need to check whether the predictions are physically meaningful and preserve desired physical quantities, such as Turbulence Kinetic Energy, Divergence and Energy Spectrum. Therefore, we include a set of additional metrics for evaluation.

**Root Mean Square Error** We calculate the RMSE of all predicted values from the ground truth for each pixel, $\sqrt{\sum_{i=1}^{N}(\hat{\boldsymbol{w_i}} - \boldsymbol{w_i})^2/N}$.

**Divergence** Since we investigate incompressible turbulent flows in this work, which means the divergence, $\nabla \cdot \mathbf{w}$, at each pixel should be zero, we use the average of absolute divergence over all pixels at each prediction step as an additional evaluation metric.

**Turbulence Kinetic Energy** In fluid dynamics, turbulence kinetic energy is the mean kinetic energy per unit mass associated with eddies in turbulent flow. Physically, the turbulence kinetic energy is characterised by measured root mean square velocity fluctuations, $(\overline{(u')^2} + \overline{(v')^2})/2$, where $\overline{(u')^2} = \frac{1}{T}\sum_{t=0}^{T}(u(t) - \bar{u})^2$ and $t$ is the time step. We calculate the turbulence kinetic energy for each predicted sample of 60 velocity fields.

**Energy Spectrum** The energy spectrum of turbulence, $E(k)$, is related to the mean turbulence kinetic energy as $\int_0^\infty E(k)dk = (\overline{(u')^2} + \overline{(v')^2})/2$. $k$ is the wavenumber, the spatial frequency in 2D Fourier domain. We calculate the Energy Spectrum on the Fourier transformation of the Turbulence Kinetic Energy fields. The large eddies have low wavenumbers and the small eddies correspond to high wavenumbers. The spectrum tells how much kinetic energy is contained in eddies with wavenumber $k$.

## 6 RESULTS

Figure 4 shows the growth of RMSE with prediction horizon up to 60 time steps ahead. `TF-Net` consistently outperforms all baselines, and constraining it with divergence free regularizer can further improve the performance. We also found `DHPM` is able to overfit the training set but performs

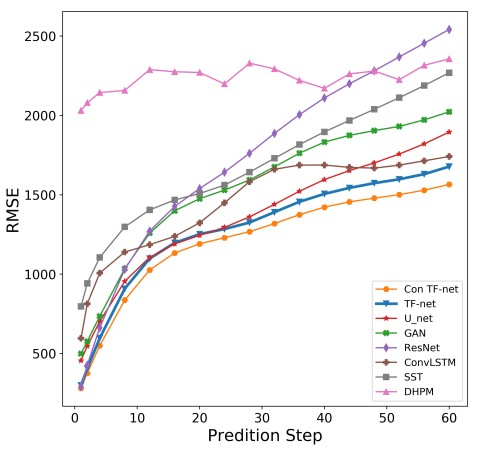

Figure 4: Root mean square errors of different models' predictions at varying forecasting horizon

Figure 5: Mean absolute divergence of different models' predictions at varying forecasting horizon

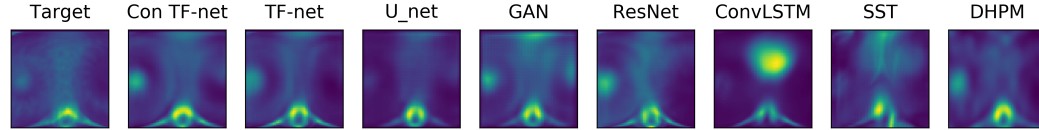

Figure 6: Turbulence kinetic energy of all models' predictions at the leftmost square field in the original rectangular field with respect to the target.

poorly when tested outside of the training domain. Neither Dropout nor regularization techniques can improve its performance. Also, the warping scheme of the Emmanuel de Bezenac (2018) relies on the simplified linear assumption, which was too limiting for our non-linear problem.

Figure 5 shows the averages of absolute divergence over all pixels at each prediction step. `TF-Net` has lower divergence than other models even without additional divergence free constraint for varying prediction step. It is worth mentioning that there is a subtle trade-off between RMSE and divergence. Even though explicitly constraining model with the divergence-free regularizer can reduce the divergence of the model predictions, it also has the side effect of smoothing out the small scale eddies, which results in a larger RMSE.

Figure 6 displays the turbulence kinetic energy fields of all models' predictions at the leftmost square field in the original rectangular field. Figure 7 shows the energy spectrum of our model and two best baseline at the leftmost square sub-field. We also convert square predicted images back to the big rectangular ones and calculate the Energy Spectrum on the entire domain, which can be found in Figure 10 in the appendix. While the turbulence kinetic energy of `TF-Net`, `U-net` and `ResNet` appear

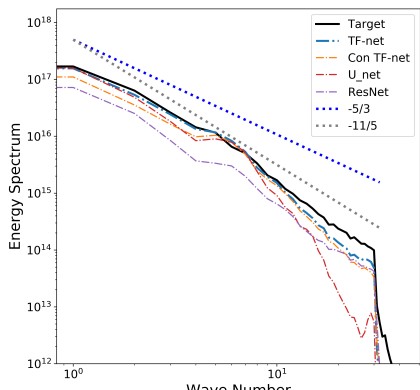

Figure 7: The Energy Spectrum of `TF-Net`, `U-net` and `ResNet` on the leftmost square sub-region.

to be similar in Figure 6, however, from the energy spectrum in Figure 7 and Figure 10, we can see that `TF-Net` predictions are in fact much closer to the target. Extra divergence free constraint does not affect the energy spectrum of predictions. Thus, unlike other models, `TF-Net` is able to generate predictions that are physically consistent with the ground truth.

Figure 8 shows the ground truth and the predicted $u$ velocity fields from all models from time step 0 to 60. We also provide videos of predictions by `TF-Net` and several best baselines in https://www.youtube.com/watch?v=sLuVGIuEE9A and https://www.youtube.com/watch?v=VMeYHID5LL8, respectively. We see that the predictions by our `TF-Net` model

are the closest to the target based on the shape and the frequency of the motions. `GAN` is able to generate flows with fine-grained details but physics are not captured in a correct way, which also shows the benefit of developing hybrid models with embedded physics knowledge. `U-net` is the best performing data-driven video prediction baseline. N. Thuerey (2019) also found the U-net architecture is quite effective in modeling dynamics flows.

We also performed an additional ablation study of TF-net to understand each component of TF-net investigate whether the TF-net has actually learned the flow with different scales. The video, `https://www.youtube.com/watch?v=ysdrMUfdhe0`, includes the predictions of TF-net, and the outputs of each small U-net while the other two encoders are zeroed out. We can see that the outputs of each small u-net are the flow with different scales. During Inference, we applied the trained TF-net to the entire input domain instead of square sub-regions. We observed that the boundaries between square sub-regions in the previous videos have disappeared. We also did the same experiments on an additional dataset (Rayleigh number = $10^5$). `TF-Net` still consistently outperforms the best two baselines, `U-net` and `ResNet`, based on all four evaluation metrics. The results are shown in in Figure 12 in the appendix.

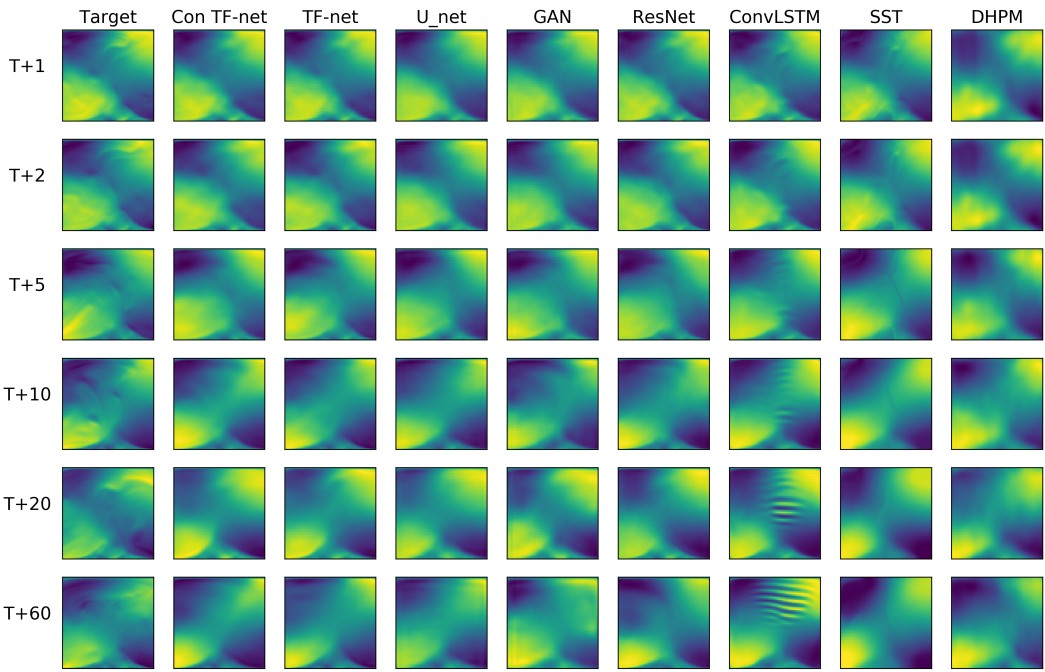

Figure 8: Ground truth and predicted $u$ velocities by all models. From left to right, constrained `TF-Net`, `TF-Net` and all the baselines. From top to bottom, predictions from time $T+1$ to $T+60$ (suppose $T$ is the time step of the last input frame).

## 7 DISCUSSION AND FUTURE WORK

We have presented a novel hybrid deep learning model, `TF-Net`, that unifies representation learning and turbulence simulation techniques. `TF-Net` exploits the multi-scale behavior of turbulent flows to design trainable scale-separation operators to model different ranges of scales individually. We provide exhaustive comparisons of `TF-Net` and baselines and observe significant improvement in both the prediction error and desired physical quantifies, including divergence, turbulence kinetic energy and energy spectrum. We argue that different evaluation metrics are necessary to evaluate a DL model's prediction performance for physical systems that include both accuracy and physical consistency. A key contribution of this work is the skillful combination of state-of-the-art turbulent flow simulation paradigms with deep learning. Future work includes extending these techniques to very high-resolution predictions, 3D turbulent flows and incorporating additional physical variables to improve the accuracy and faithfulness of physically-informed deep learning models.

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

# A APPENDIX

## A.1 ADDITIONAL RESULTS

Table 1 displays the number of parameters, the best number of input frames, the best number of accumulated errors for backpropagation and training time for one epoch on 8 v-100 GPUs for each model. We can conclude that our model has significantly smaller number of parameters than most baselines yet achieves the best performance. About 25 historic images are enough for deep learning models to generate reasonable predictions, and ConvLSTM require large memory and training time, especially when the number of historic input frames is large. Additionally, Table 2 displays the hyper-parameters tuning range of models.

| Models | TF-net | U_net | GAN | ResNet | ConvLSTM | SST | DHPM |
|---|---|---|---|---|---|---|---|
| **#params(10^6)** | 15.9 | 25.0 | 26.1 | 21.2 | 11.8 | 49.9 | 2.12 |
| **input length** | 25 | 25 | 24 | 26 | 27 | 23 | \ |
| **#accumulated errors** | 4 | 6 | 5 | 5 | 4 | 5 | \ |
| **time for one epoch(min)** | 0.39 | 0.57 | 0.73 | 1.68 | 45.6 | 0.95 | 4.591 |

Table 1: The number of parameters, the best number of input frames, the best number of accumulated errors for backpropogation and training time for one epoch on 8 v-100 GPUs for each model.

| Hyper-parameters Tuning Range | | | | | | | |
|---|---|---|---|---|---|---|---|
| Learning rate | Batch Size | #Accumulated errors for backpropogation | #Input frames | Moving average window size (TF-net) | Size of the spatial filter (TF-net) | #Layers (ConvLSTM) | Hidden Dimension (ConvLSTM) |
| 1e-1 ∼1e-6 | 16 ∼128 | 1 ∼10 | 1 ∼30 | 2∼10 | 3∼9 | 1 ∼5 | 32 ∼512 |

Table 2: Hyper-parameters tuning ranges

Figure 9 shows the ground truth and the predicted $v$ velocity fields over 60 time steps. Similar to the $u$ velocity predictions, we observe that the predictions from `TF-Net` are the closest to the target. `U-net` and `GAN` generate smooth predictions and miss the details of small scale motion. There is still room for improvement in long-term prediction for all the models.

We converted square predicted images back to the big rectangular ones and calculated the Energy Spectrum on the entire domain, as shown in Figure 10. we can see that `TF-Net` predictions are in fact much closer to the target on small wavenumbers and more stable on large wavenumbers.

We visualized the learned filters in Figure 11. We only found two types of spatial and temporal filters from all trained `TF-Net` models, with and without the divergence regularizer. The meaning of these learned filters are yet to be explored.

We also performed the same experiments on an additional dataset (Rayleigh number =$10^5$). `TF-Net` still consistently outperforms the best two baselines, `U-net` and `ResNet`, based on all four evaluation metrics. The results are shown in Figure 12.

## A.2 IMPLEMENTATION DETAILS

We adapt `SST` Emmanuel de Bezenac (2018) to model non-linear turbulent flow. `SST` successfully infused a deep learning model into the solution of the linear energy equation to predict sea surface temperature. If we make the assumption that the advection term $(\mathbf{w} \cdot \nabla)u$ in the momentum equation is a linear term $(\mathbf{c} \cdot \nabla)u$, where $\mathbf{c}$ is unknown, then we can use two separate models to predict $u$ and $v$, and the inputs of both parts are the same stacked $u$ and $v$ from previous time steps.

For `DHPM` (Raissi, 2018), we approximate both the velocity field two 6-layer neural networks with 512 neurons per hidden layer and use and a 4-layer neural networks with 512 neurons per hidden layer to represent pressure $p$ and an non-homogeneous term $f$ that encapsulates the influence of temperature and viscosity. During the training, we make sure the outputs of these neural networks satisfy the continuity and momentum equations. It is worth mentioning that the `DHPM` model is supposed to be trained twice, first on the training set then on the initial and boundary conditions of the test set. This model can be formulated as below.

$$Loss = \|\boldsymbol{w} - \hat{\boldsymbol{w}}\| + \|\nabla \cdot \hat{\boldsymbol{w}}\| + \|\hat{\boldsymbol{w}}_t + (\hat{\boldsymbol{w}} \cdot \nabla)\hat{\boldsymbol{w}} - \nu\nabla^2\hat{\boldsymbol{w}} - f\|$$

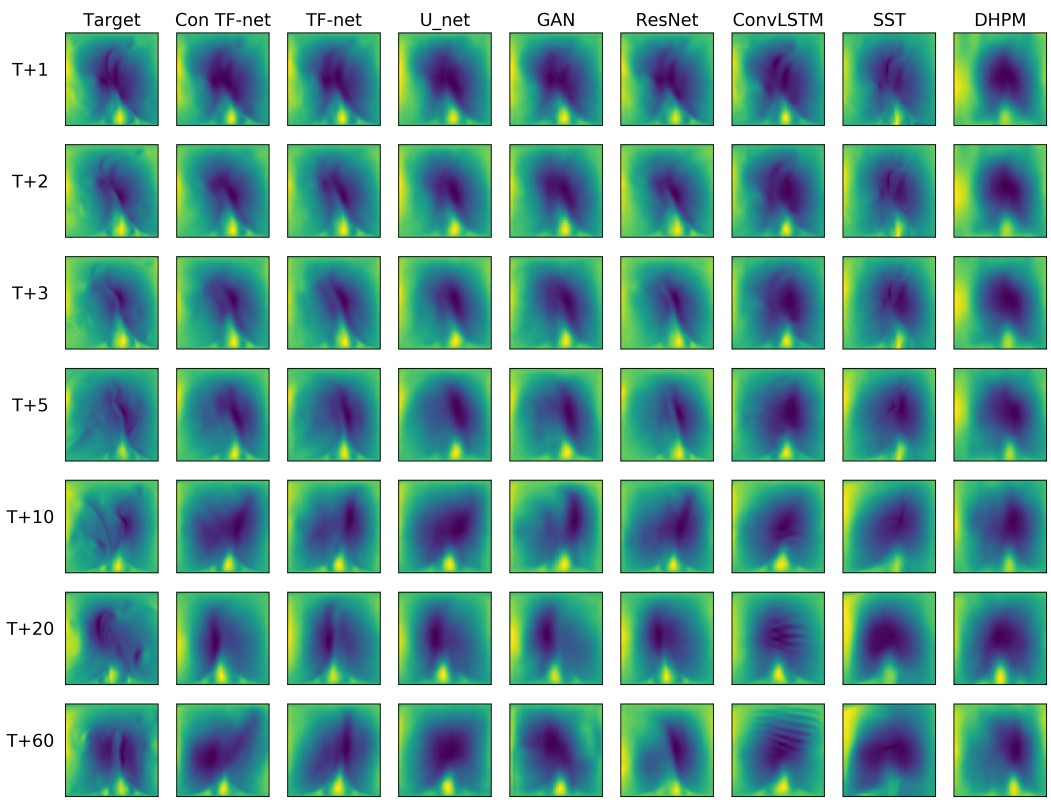

Figure 9: Ground truth and predicted $v$ velocities by models, suppose $T$ is the time step of the last input image.

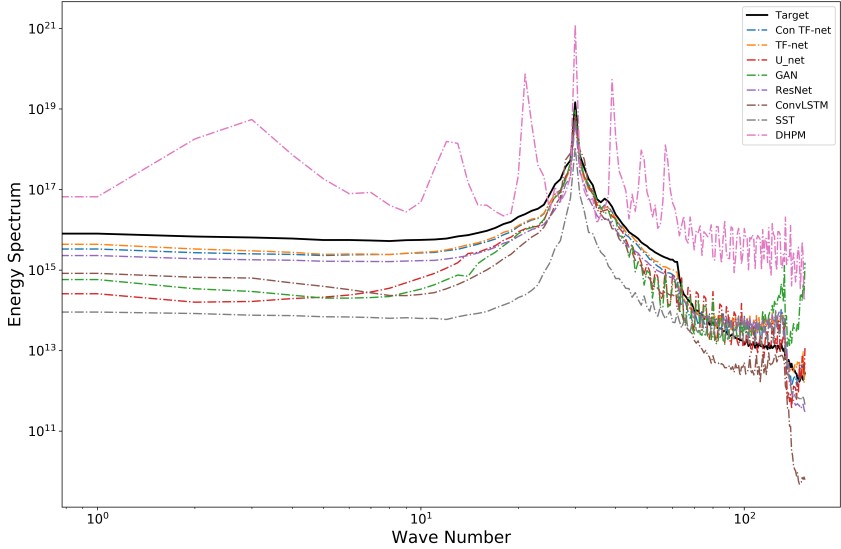

Figure 10: Energy Spectrums of all models' predictions on the entire rectangular domain.

Where $\hat{\boldsymbol{w}} = NN(x, y, t)$, and $f = NN(x, y, t, \hat{u}, \hat{v}, \hat{u}_x, \hat{v}_x, \hat{u}_y, \hat{v}_y, \hat{u}_{xx}, \hat{v}_{xx}, \hat{u}_{yy}, \hat{v}_{yy})$

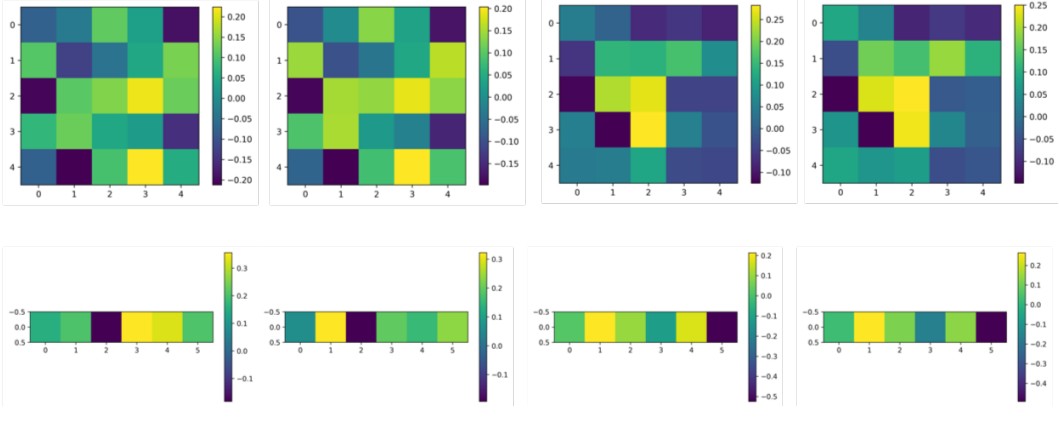

Figure 11: Learned spatial and temporal filters in `TF-Net`

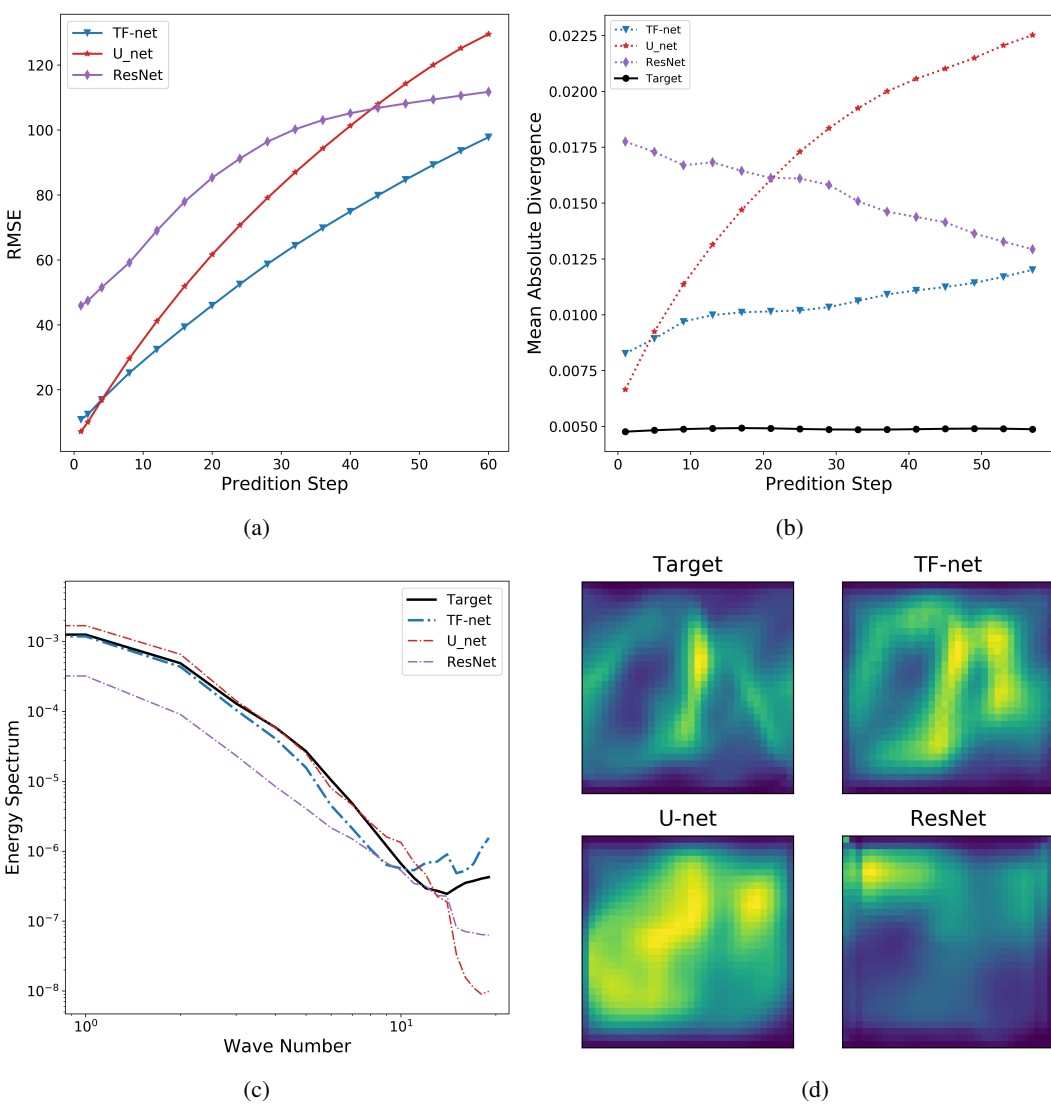

Figure 12: The performances of TF-net, U-net and ResNet on an additional dataset(Ra = 10000). (a): Root mean square errors of different models' predictions at varying forecasting horizon, (b): Mean absolute divergence of models' predictions at varying forecasting horizon, (c): The Energy Spectrums on the entire domain, (d): Turbulence kinetic energy fields of three models' predictions.

