# OpenReview forum: "Towards Physics-informed Deep Learning for Turbulent Flow Prediction"
_ICLR.cc/2020/Conference — Reject_

### Official Review · AnonReviewer3 · 2019-10-22
**Official Blind Review #3**

**Rating:** 6

**Review:**

This work targets learning flow fields for two-dimensional Rayleigh-Benard convections inspired by hybrid RANS-LES turbulence models.

Internally, the approach employs three U-nets which are each trained to predict a 32 dimensional feature space from three variants of filtered velocity fields. These three feature spaces are (probably) concatenated and translated to an output velocity via a fourth U-net. While the overall architecture is intuitive, details are missing in the text.

I also assume that the network outputs w for timestep t+1, which is the sole input for the next evaluation of the network? A regular flow solver is not used in conjunction with this network, but the results (i.e. sequences) shown are purely inferred by the network? (I guess G2 uses multiple frames - what is used for T here, btw.?)

These details, plus specifics of each layer should be written out (e.g. in the appendix), together with operations such as the merging of the three encoder outputs to make the work reproduicble. I hope the authors can also clarify these points in the rebuttal.

For a future version, I'd also recommend to rephrase equations (2) and (3). The split into w, w-bar and w-tilde is not compatible with figure 2. The figure uses the split from equations (6,7), so it would be good to make this clear via the notation.

As training data, the model uses single RBC data set produced with a lattice boltzmann method. It's a pity only a single case is shown, as the method claims to learn a general turbulence model. Do the authors have a second data set on which they could demonstrate the method? This would show help to show generality of the approach.

The RBC test case is used to train a nice range of different methods, from a simple ResNet to approaches from previous work, and the results are evaluated with a good range of turbulence metrics. These evaluations show nice improvements, e.g., the RMSEs over time in figure 4 are consistently lower. Unfortunately, it's not made clear which data is evaluated - is this a single training case, or e.g. averaged for the whole test set? Likewise for figure 6 and 7.

Very minor, but I'd recommend to rephrase the last sentence of the fluid animation discussion. Those works are part of computer science, which arguably also counts as science.

Overall, I found the split into temporally and spatially filtered components of the flow field is an interesting one, and it's nice to see how well this seems to work. The paper certainly does not aim for new insight for deep learning methods in general, but provides an interesting application for turbulent flows that is evaluated with a nice amount of detail. If the authors can address the unclear points mentioned above, and maybe include a second test case that is evaluated on a subset of the different models, I think this paper could be included in the ICLR program.


**Experience Assessment:**

I have published in this field for several years.

**Review Assessment: Checking Correctness Of Derivations And Theory:**

I assessed the sensibility of the derivations and theory.

**Review Assessment: Checking Correctness Of Experiments:**

I carefully checked the experiments.

**Review Assessment: Thoroughness In Paper Reading:**

I read the paper thoroughly.

---

> ### Author Response · Authors · 2019-11-14
> **Answers**
>
> Thanks so much for your detailed comments.
>
> >>> I also assume that the network outputs w for timestep t+1, which is the sole input for the next evaluation of the network? A regular flow solver is not used in conjunction with this network, but the results (i.e. sequences) shown are purely inferred by the network? (I guess G2 uses multiple frames - what is used for T here, btw.?)
> Answer: Yes, for all CNN based models included in our paper, we make predictions in an autoregressive manner. Suppose V is the Velocity field, N is the number of input frames and T is the timestamp for the last input frame. The models took N initial frames V(T-N+1,..,T) as input to make the first prediction for the next step(T+1), then we feed the prediction at (T+1) back to the input V(T-N+2,..,T+1) to make predictions for time step T+2, and repeat for 60 times.  \bar{w} is a weighted average, T here is the moving average window size.
>
> >>> These details, plus specifics of each layer should be written out (e.g. in the appendix), together with operations such as the merging of the three encoder outputs to make the work reproducible. I hope the authors can also clarify these points in the rebuttal.
> Answer:  The detailed numbers of each layer are shown in Figure 2. The merging of the three encoder outputs is simply summation. We will open source our implementation in the final version.
>
> >>> For a future version, I'd also recommend to rephrase equations (2) and (3). The split into w, w-bar and w-tilde is not compatible with figure 2. The figure uses the split from equations (6,7), so it would be good to make this clear via the notation.
> Answer: Thanks for pointing it out. We modified those two equations in the updated version.
>
> >>> Unfortunately, it's not made clear which data is evaluated - is this a single training case, or e.g. averaged for the whole test set? Likewise for figure 6 and 7.
> Answer: All evaluation metrics are averaged over the whole test set.

---

### Official Review · AnonReviewer2 · 2019-10-24
**Official Blind Review #2**

**Rating:** 3

**Review:**

## Summary

The authors propose use learned spatio-temporal filtering and a convolutional model to predict the behavior of a turbulent fluid flow. Turbulent flow is a very difficult problem, of great interest for engineers and physical scientists, so the topic of the paper is certainly compelling.

This paper has several significant issues. The baselines the authors compare to are quite weak. Many of them were designed for other purposes, such as video prediction. The authors claim significant improvements ("64.2% reduction in divergence") that I believe are calculated in a (unintentionally) misleading way. In fact, I think they missed the most important baseline---the ground truth simulation itself. Unless they can argue that the learned model is superior to the classical simulation in a significant way, it is hard to see the benefit of using something like TF-NET.

## Specific Comments

* Page 2: The claim "64.2% reduction in flow divergence, compared to the best baseline" seems misleading. In Figure 5, the constrained TF-NET is as low as ~590 but the ResNet is between ~610 and ~810. I am guessing that the authors meant a 64.2% reduction in *difference* from the Target model, but this should be clarified.
* Page 4: Is 'T' supposed to appear in the denominator of Equation 3? I would have expected this to be a normalizing factor resulting from integrating the filter G over the whole domain.
* Page 4: I would expect filters used for LES are symmetric. Are any symmetry requirements being enforced on the learned filters?
* Page 4: It's not clear what TF-NET is outputting. I would expect it to output the time derivative of the velocity field, but this should be spelled out explicitly.
* Page 5: Since you are using incompressible Navier-Stokes equations, I don't think the Mach number is relevant. IIUC this is only relevant for the propogation of shock waves in compressible flows.
* Page 7: It looks as if the Target model has significant divergence. Why is this? Should we expect this much divergence in the ground truth data?
* Page 8: In homogeneous isotropic turbulent flow, the energy spectrum is governed by Kolmogorov, so we know what the spectrum should look like. Is there an analytic result for RB convection? If so, could you include that on the plot?
* Page 8: The lines for U-Net and TF-NET are nearly indistinguishable in Figure 7. Could you change them?
* Page 8: It seems like it would be worth including the ResNet in the energy spectrum plots as well.
* What did the learned spatial and temporal filters look like? How do they compare to typical 'hand-chosen' filters?
* IIUC, the training, validation and test data all have identical Rayleigh number. Does the learned model generalize to higher/lower energy? This seems critical to making this sort of model useful.
* The paper doesn't describe a tuning process for any of the models. I would expect this to lead to significant improvement. In particular, how do the models' performance change as the weight on the divergence loss term is increased?

### Baselines
* I think the objective should be to show that TF-NET is superior to the ground-truth method in some way. Can it match the results of the ground truth simulator but do it faster, or with fewer resources?
* The authors say "we compare our model with a series of state-of-the-art baselines for turbulent flow prediction." However, most of these models are intended for video prediction or other tasks unrelated to turbulent flows. I wouldn't expect any of the baselines considered to do perform well in this context.
* Since TF-NET gets the benefit of a loss related to divergence, I would have expected the other architectures to get the same treatment. In fact, without this extra loss term, the ResNet architecture is competitive with TF-NET.

**Experience Assessment:**

I have read many papers in this area.

**Review Assessment: Checking Correctness Of Derivations And Theory:**

I assessed the sensibility of the derivations and theory.

**Review Assessment: Checking Correctness Of Experiments:**

I assessed the sensibility of the experiments.

**Review Assessment: Thoroughness In Paper Reading:**

I read the paper thoroughly.

---

> ### Author Response · Authors · 2019-11-14
> **Answers**
>
> Thanks for your detailed comments and constructive feedback.
>
> >>> Page 2: The claim "64.2% reduction in flow divergence, compared to the best baseline" seems misleading. In Figure 5, the constrained TF-NET is as low as ~590 but the ResNet is between ~610 and ~810. I am guessing that the authors meant a 64.2% reduction in *difference* from the Target model, but this should be clarified.
>      Answer: Thank you, we have clarified it in the updated version.
>
> >>> Page 4: Is 'T' supposed to appear in the denominator of Equation 3? I would have expected this to be a normalizing factor resulting from integrating the filter G over the whole domain.
>      Answer: No, that’s a typo. We have fixed it.
>
> >>> Page 4: I would expect filters used for LES are symmetric. Are any symmetry requirements being enforced on the learned filters?
>      Answer: We did not explicitly enforce symmetry in the LES filters. Such symmetry is learned implicitly from data, see figure 12 in the appendix.
>
> >>> Page 4: It's not clear what TF-NET is outputting. I would expect it to output the time derivative of the velocity field, but this should be spelled out explicitly.
>      Answer: TF-net outputs two channels, the velocity along the x direction and the velocity along the y direction.
>
> >>> Page 5: Since you are using incompressible Navier-Stokes equations, I don't think the Mach number is relevant. IIUC this is only relevant for the propogation of shock waves in compressible flows.
> Answer: We emphasize that the maximum Mach number anywhere in this flow is less than 0.1, which justifies the Boussinesq approximation, which is relevant to the RBC system. Hence we ignore acoustic terms and use the incompressible Navier-Stokes equations with density appearing only when in the body force term. If the Mach number was larger (0.3 and above) then the compressible Navier-Stokes equations would have to be used.
>
> >>> Page 7: It looks as if the Target model has significant divergence. Why is this? Should we expect this much divergence in the ground truth data?
>      Answer: This is because when we calculate the divergence, we used second-order image derivatives, which is an approximation of the actual spatial derivatives. Also, the images have lower resolution than the ones during the simulation.
>
> >>> Page 8: In homogeneous isotropic turbulent flow, the energy spectrum is governed by Kolmogorov, so we know what the spectrum should look like. Is there an analytic result for RB convection? If so, could you include that on the plot?
>      Answer: We added the power spectrum plot of the predictions on the leftmost square sub-region in section 6, Figure 7. As in homogeneous isotropic turbulent flow, the energy spectrum is governed by the Kolmogorov-Obukhov scaling law (-5/3 slope) in the inertial sub-range and by the Bolgiano-Obukhov scaling law (-11/5 slope) in the buoyancy sub-range.
>
> >>> Page 8: The lines for U-Net and TF-NET are nearly indistinguishable in Figure 7. Could you change them?
>      Answer: Sure
>
> >>>  Page 8: It seems like it would be worth including the ResNet in the energy spectrum plots as well.
>      Answer: The energy spectrum plots for all models are in Figure 10 in the appendix. We also added the power spectrum plot of the predictions on the leftmost square sub-region in the updated version which includes ResNet.
>
> >>> What did the learned spatial and temporal filters look like? How do they compare to typical 'hand-chosen' filters?
>      Answer: We only found two types of spatial and temporal filters from all trained models, with and without the divergence regularizer. The meaning of these learned filters is yet to be explored. We included those filters in the appendix Figure 11.
>
> >>> IIUC, the training, validation and test data all have identical Rayleigh number. Does the learned model generalize to higher/lower energy? This seems critical to making this sort of model useful.
>      Answer: We performed additional experiments on a new dataset (Rayleigh number = 10^5) to validate the generalization of our method. TF-net still consistently outperforms the best two baselines, U-net and ResNet, across all four evaluation metrics. The results can be found in Appendix Figure 12.
>
> * The paper doesn't describe a tuning process for any of the models. I would expect this to lead to significant improvement. In particular, how do the models' performance change as the weight on the divergence loss term is increased?
>      Answer: We performed an exhaustive grid search for hyper-parameters for all models included in our paper. See Appendix Table 2 for the tuning range. Increasing the weight on the divergence-free regularizer can reduce the divergence of the model predictions but it also has the side effect of smoothing out the small scale eddies, which results in a larger RMSE.

---

> > ### Author Response · Authors · 2019-11-14
> > **Answers**
> >
> > ### Baselines
> > >>> I think the objective should be to show that TF-NET is superior to the ground-truth method in some way. Can it match the results of the ground truth simulator but do it faster, or with fewer resources?
> >      Answer: Inference from a well-trained dl model basically need no time. The numerical simulation requires much time.
> >
> > >>> The authors say "we compare our model with a series of state-of-the-art baselines for turbulent flow prediction." However, most of these models are intended for video prediction or other tasks unrelated to turbulent flows. I wouldn't expect any of the baselines considered to perform well in this context.
> >      Answer: Two of our baselines, DHPM and SST,  are already the state-of-the-art for physics informed DL. To the best of our knowledge, other than the numerical methods, there are not many DL works for turbulence prediction, which also highlights the novelty of our work.
> >
> > >>> Since TF-NET gets the benefit of a loss related to divergence, I would have expected the other architectures to get the same treatment. In fact, without this extra loss term, the ResNet architecture is competitive with TF-NET.
> >      Answer: We respectfully disagree. In fact, ResNet has much larger RMSE than TF-net for varying forecasting horizons in Figure 4. And from the video, predictions from ResNet miss lots of small eddies and have much slower motions compared with the target and TF-net. We don’t think the ResNet architecture is competitive with TF-NET. Even without divergence-free regularizer, TF-Net already outperforms all other baselines across all four evaluation metrics.

---

> > > ### Comment · AnonReviewer2 · 2019-11-15
> > > **Thanks for addressing my comments**
> > >
> > > The authors did a nice job of addressing my comments (and those of the other reviewers). The energy spectrum plot (Fig 7) is a particularly nice addition.
> > >
> > > I still think the authors should make a better argument for why we would use TF-Net instead of numerical simulation. In their response, they say "Inference from a well-trained dl model basically need no time. The numerical simulation requires much time." If this is so, it would be great to include a table of wall clock times for each method.
> > >
> > > The topic of the paper is really interesting, and the authors have really improved the paper based on reviewers' comments. As a result, I would be happy to raise my rating to a 5. If the other reviewers believe the paper should be accepted, I wouldn't object.

---

### Official Review · AnonReviewer1 · 2019-10-25
**Official Blind Review #1**

**Rating:** 6

**Review:**

This paper shows a method that combines a convolutional neural network with a multi-scale physical computational fluid dynamics (CFD), in the scope of predicting turbulent flows. The authors proposed a new network, TF-Net, that is based on a multi-scale CFD: a temporal and a spatial filters are learned, prior to 3 separate 'encoder' networks, that are then grouped in a unique 'decoder' part. The method is tested on a synthetic example, showing interesting results and comparing with a large set of baselines.

Overall, I found the paper interesting in that it really starts from modeling and CFD aspects, in order to derive a neural network architecture. However, I am a not totally convinced since (i) the authors said that their goal is to emulate numerical simulations given 'noiseless observations': how can this be used on a real case then? and (ii) the developed (complex) architecture is still not sufficient to constrain the physical problem, as they had to manually include a regularization term (on the divergence) in the loss: what is the purpose of the 'physically-based network' then, and how can this method be extended to other dynamics?

The paper is quite well written, the method compared with a lot of state-of-the-art methods, and performs well on this noise-free problem.

Remarks/ questions:
- In this paper, we suppose that the equations are known. What if it is not the case (at least partially)? Would it still be applicable? In the reality, there are other components of the dynamics that might be not fully understood, some noise, ect...
- As said previously, and also as you mentioned in the introduction, the literature 'focused [...] on regularization being ad-hoc and difficult to tune'. But in the end, you also need a regularization term in your loss to prevent a non-zero divergence. So what is different then?
- The total size of the image is a rectangle composed of 7 square sub-regions, each of them being used separately as the input of the network. But in the result, we do see clearly some boundary effects. Would it be possible to also learn the 'intermediate' squares, in between the n. 1 and the n. 2 for example, and then to reconstruct the full picture with less boundary effects? Otherwise, I see a limitation: since your work is based on known equations without noise, the goal is only the speedup of the computation. But if the results are too degraded, I don't see how it can be used.
- Do you think that your model is able to learn the underlying physics? If so, do we have access to the latent variables, that would be the state of your system (see Learning Dynamical Systems from Partial Observations, I. Ayed, https://arxiv.org/pdf/1902.11136.pdf)? Such as p and T.
- Similarly, is there a way to understand each component of your network? Such as (i) represent the spatial and temporal filters learned; (ii) understand the 3 U-nets: to what inputs do they respond, what are their respective outputs (before the 'decoder')? Is it doing what we wanted, are they all useful?
- Since you are not using recurrent networks, why don't you automatically predict at 60 frames from the start?

Small remarks:
- 'quantifies': few times, did not not mean 'quantities'?
- 'N. Thuery (2019) also found the U-net architecture is quite' -> a 'that' is missing

**Experience Assessment:**

I have read many papers in this area.

**Review Assessment: Checking Correctness Of Derivations And Theory:**

I assessed the sensibility of the derivations and theory.

**Review Assessment: Checking Correctness Of Experiments:**

I carefully checked the experiments.

**Review Assessment: Thoroughness In Paper Reading:**

I read the paper at least twice and used my best judgement in assessing the paper.

---

> ### Author Response · Authors · 2019-11-14
> **Answers**
>
> Thanks for your detailed comments and constructive feedback.
>
> >>> In this paper, we suppose that the equations are known. What if it is not the case (at least partially)? Would it still be applicable? In reality, there are other components of the dynamics that might be not fully understood, some noise, ect…
>       Answer: Most fluids are governed by Navier-stokes equations. TF-Net is a multi-scale model based on the CFD filters, and hence is applicable to most fluids. TF-net does not assume the equations are known nor does it uses the equations. We are only using the general notion of multiscale dynamics to create this template of 3 networks (for slow, medium and fast scales), which is true, generally speaking, for a wide range of physical systems.
>
> >>> As said previously, and also as you mentioned in the introduction, the literature 'focused [...] on regularization being ad-hoc and difficult to tune'. But in the end, you also need a regularization term in your loss to prevent a non-zero divergence. So what is the difference then?
>       Answer: Even without divergence-free regularizer, TF-Net already outperforms all other baselines across all four evaluation metrics.  The unconstrained TF-Net is comparable with the constrained TF-Net. We have experimented with many regularizers. Only the divergence-free regularizer can improve RMSEs but still has a side effect of smoothing out small eddies. We reported the results with regularizers merely as an ablative study. Our main contribution is TF-net not the regularizer.
>
> >>>The total size of the image is a rectangle composed of 7 square sub-regions, each of them being used separately as the input of the network. But in the result, we do see clearly some boundary effects. Would it be possible to also learn the 'intermediate' squares, in between the n. 1 and the n. 2 for example, and then to reconstruct the full picture with less boundary effects? Otherwise, I see a limitation: since your work is based on known equations without noise, the goal is only the speedup of the computation. But if the results are too degraded, I don't see how it can be used.
>        Answer:  During Inference, we applied the trained TF-net to the entire input domain instead of square sub-regions. We observed that the boundaries between square sub-regions in the previous videos have disappeared. The new video can be found here https://www.youtube.com/watch?v=ysdrMUfdhe0, which includes the predictions of TF-net, and the outputs of each small U-net while the other two encoders are zeroed out.
>
> >>> Do you think that your model is able to learn the underlying physics? If so, do we have access to the latent variables, that would be the state of your system (see Learning Dynamical Systems from Partial Observations, I. Ayed, https://arxiv.org/pdf/1902.11136.pdf)? Such as p and T.
>        Answer: Our work implicitly marginalizes out latent variables (p and T). To access those, we would need the initial full state of the system as in I. Ayed paper. We are happy to consider the suggested setup as a future work.
>
> >>> Similarly, is there a way to understand each component of your network? Such as (i) represent the spatial and temporal filters learned; (ii) understand the 3 U-nets: to what inputs do they respond, what are their respective outputs (before the 'decoder')? Is it doing what we wanted, are they all useful?
>        Answer: We also performed an additional ablation study of TF-net to understand whether the TF-net has actually learned the flow with different scales. The new video, https://www.youtube.com/watch?v=ysdrMUfdhe0, includes the predictions of TF-net, and the outputs of each small U-net while the other two encoders are zeroed out. We can see that the outputs of each small U-net are indeed the flow with different scales. The learned temporal and spatial filters are shown in Figure 11 in the appendix.
>
> >>> Since you are not using recurrent networks, why don't you automatically predict at 60 frames from the start?
>         Answer: Predicting in an autoregressive manner allows us to forecast arbitrary number of steps head.  We have also tried directly predicting 60 frames for all models but the predictions are worse. In fact, one of our baselines, ConvLSTM, involves LSTM operations, which is really time-consuming and didn’t perform well.

---

### Author Response · Authors · 2019-10-30
**Ablation study**

We also performed an additional ablation study of TF-net to investigate whether the TF-net has actually learned the flow with different scales. The new video, https://www.youtube.com/watch?v=ysdrMUfdhe0, includes the predictions of TF-net, and the outputs of each small U-net while the other two encoders are zeroed out.

During Inference, we applied the trained TF-net to the entire input domain instead of square sub-regions. We observed that the boundaries between square sub-regions in the previous videos have disappeared.

---

### Author Response · Authors · 2019-11-14
**General Response**

We thank all reviewers for the detailed comments and constructive feedback.

Updates since the rebuttal:
>>> We performed experiments on a new dataset (Rayleigh number = 10^5) to validate the generalization of our method. We observe that TF-net consistently outperforms the best two baselines, U-net and ResNet, across all four evaluation metrics. The results can be found in Appendix Figure 12.
>>> We included the hyper-parameter tuning ranges in table 2 in the appendix.
>>> We added the power spectrum plot and the theoretical curve for the leftmost square sub-region in section 6  Figure 7 of the updated version.
>>> We added an ablation study of TF-net to understand whether each component of TF-net has actually learned the flow with different scales. The new video, https://www.youtube.com/watch?v=ysdrMUfdhe0, includes the predictions of TF-net, and the outputs of each small U-net while the other two encoders are zeroed out. We can see that the outputs of each small U-net indeed correspond to the flow with different scales.
>>> During Inference, we tried applying the trained TF-net to the entire input domain instead of the square sub-regions. We observe that the boundaries between square sub-regions in the previous videos have disappeared.

---

### Decision · Program_Chairs · 2019-12-19

**Decision:**

Reject

**Comment:**

The reviewers all agree that this is an interesting paper with good results. The authors' rebuttal response was very helpful. However, given the competitiveness of the submissions this year, the submission did not make it. We encourage the authors to resubmit the work including the new results obtained during the rebuttal.